# Local and Central Evaluation of HER2 Positivity and Clinical Outcome in Advanced Gastric and Gastroesophageal Cancer—Results from the AGMT GASTRIC-5 Registry

**DOI:** 10.3390/jcm9040935

**Published:** 2020-03-29

**Authors:** Florian Huemer, Lukas Weiss, Peter Regitnig, Thomas Winder, Bernd Hartmann, Josef Thaler, Gudrun Piringer, Clemens A. Schmitt, Wolfgang Eisterer, Hannes Gänzer, Alois Wüstner, Johannes Andel, Björn Jagdt, Hanno Ulmer, Richard Greil, Ewald Wöll

**Affiliations:** 1Department of Internal Medicine III with Haematology, Medical Oncology, Haemostaseology, Infectiology and Rheumatology, Oncologic Center, Paracelsus Medical University, Salzburg 5020, Austria; f.huemer@salk.at (F.H.); lu.weiss@salk.at (L.W.); r.greil@salk.at (R.G.); 2Salzburg Cancer Research Institute—Center for Clinical Cancer and Immunology Trials (SCRI-CCCIT); Salzburg 5020, Austria; 3Diagnostic and Research Institute of Pathology, Medical University of Graz, Graz 8010, Austria; peter.regitnig@medunigraz.at; 4Department of Internal Medicine, LKH Feldkirch, Feldkirch 6800, Austria; thomas.winder@vlkh.net (T.W.); bernd.hartmann@lkhf.at (B.H.); 5Department of Internal Medicine IV, Hematology and Oncology, Klinikum Wels-Grieskirchen, Wels 4600, Austria; josef.thaler@klinikum-wegr.at (J.T.); gudrun.piringer@klinikum-wegr.at (G.P.); 6Kepler Universitätsklinikum, Linz 4020, Austria; clemens.schmitt@kepleruniklinikum.at; 7Medical University Hospital Innsbruck, Innsbruck 6020, Austria; wolfgang.eisterer@kabeg.at; 8General Hospital Hall, Hall 6060, Austria; hannes.gaenzer@kh-schwaz.at; 9General Hospital Hohenems, Hohenems 6845, Austria; alois.wuestner@gmx.at; 10Landeskrankenhaus Steyr, Steyr 4400, Austria; johannes.andel@ooeg.at; 11Krankenhaus der Barmherzigen Schwestern Ried, Ried 4910, Austria; bjoern.jagdt@bhs.at; 12Department of Medical Statistics, Informatics and Health Economics, Innsbruck Medical University, Innsbruck 6020, Austria; hanno.ulmer@i-med.ac.at; 13Cancer Cluster Salzburg, Salzburg 5020, Austria; 14St. Vinzenz Krankenhaus Betriebs GmbH, Zams 6511, Austria

**Keywords:** gastric cancer, gastroesophageal cancer, HER2, concordance, discordance

## Abstract

Trastuzumab in combination with a platinum and fluorouracil is the treatment of choice for patients with advanced human epidermal growth factor receptor 2 (HER2) positive gastric cancer and gastroesophageal junction (GEJ) cancer. Pathological assessment of the HER2 status in gastric/GEJ cancer, however, still remains difficult. However, it is a crucial prerequisite for optimal treatment. The GASTRIC-5 registry was designed as an observational, multi-center research initiative comparing local and central HER2 testing. HER2 status was assessed by immunohistochemistry (IHC) and in equivocal cases (IHC score 2+) by additional in-situ hybridization. Between May 2011 and August 2018, tumor samples of 183 patients were tested in local and central pathology laboratories, respectively. Central testing revealed HER2 positivity in 38 samples (21%). Discordant HER2 results were found in 12% (22 out of 183) with locally HER2 positive/centrally HER2 negative results (9%, 17 out of 183), exceeding locally HER2 negative/centrally HER2 positive results (3%, 5 out of 183). Centrally confirmed HER2 positive patients receiving trastuzumab-based palliative first-line therapy showed a longer median overall survival compared to centrally HER2 positive patients not receiving trastuzumab (17.7 months (95% CI: 10,870–24,530) vs. 6.9 months (95% CI: 3.980–9.820), *p* = 0.016). The findings of the GASTRIC-5 registry corroborate the challenge of HER2 testing in gastric/GEJ cancer and highlight the necessity for central quality control to optimize individual treatment options. Centrally HER2 positive patients not receiving trastuzumab had the worst outcome in a Western real-world gastric/GEJ cancer cohort.

## 1. Introduction

Over the last decades, gastric cancer incidence and mortality rates have been continuously declining in Europe [1] and in the United States [2]. Human epidermal growth factor receptor 2 (HER2) positive disease shows an aggressive clinical behavior and HER2 positivity represents a negative prognostic factor in gastric/gastroesophageal junction (GEJ) cancer [3,4,5,6].

In metastatic gastric/GEJ cancer, 16–22% of samples are tested HER2 positive [7,8], and as a consequence, HER2 status testing is mandatory to guide optimal front-line systemic therapy [9,10,11]. In advanced HER2 negative disease, a median overall survival (OS) ranging from 8.6–12.5 months can be achieved with sequential palliative systemic therapy [12,13]. The phase III “Trastuzumab for Gastric Cancer” ToGA trial demonstrated a statistically significant improvement of median OS from 11.1 to 13.8 months by the addition of the HER2-targeting antibody trastuzumab to a platinum and fluorouracil (5-FU) based doublet chemotherapy in HER2 positive gastric/GEJ cancer [7], and since then has become the standard first-line regimen in HER2 positive disease [9,10]. In contrast to breast cancer, a benefit for anti-HER2 targeting therapy seen with trastuzumab in first-line treatment of HER2 positive gastric/GEJ cancer could not be reproduced with other HER2-targeting strategies: no survival benefit was seen with the antibody-drug conjugate trastuzumab emtansine (T-DM1) [14] in second-line, with the tyrosine kinase inhibitor lapatinib in first-line [15] and second-line [16], or with the addition of pertuzumab to trastuzumab and chemotherapy [17] in first-line metastatic HER2 positive gastric/GEJ cancer. Therefore, trastuzumab remains the only HER2-targeting drug approved by the European Medicines Agency (EMA) and Food and Drug Administration (FDA) in advanced HER2 positive gastric/GEJ cancer.

However, assessing the HER2 status by immunohistochemistry (IHC) and in situ hybridization (ISH) in gastric/GEJ cancer is challenging. Asian countries face high gastric/GEJ cancer incidence and mortality rates [18,19] and therefore reproducible HER2 results are an absolute necessity in clinical practice. Results from the Chinese HER-EAGLE study showed an encouraging HER2 concordance rate of 97% between local and central testing [20]. However, data on HER2 testing in Caucasian patients with advanced gastric/GEJ cancer are sparse.

The aim of the GASTRIC-5 registry was to evaluate the rate of HER2 positivity, concordance rate between local and central HER2 results as well as clinical outcome in a real-world Western advanced gastric/GEJ cancer cohort.

## 2. Experimental Section

This registry was designed by the Austrian Group for Medical Tumor Therapy (AGMT) as an observational, multi-center research initiative in Austria comparing HER2 test results obtained from the local pathology laboratory with a blinded analysis of the HER2 status obtained by a central pathology laboratory. The study was approved by the ethics committee of the Medical University of Innsbruck on 11 August 2010 (protocol number: UN4036). All patients had given their written informed consent. The primary study objective was to evaluate the rate of HER2 positive locally advanced or metastatic gastric/GEJ cancer. An assessment of concordance and discordance rates of HER2 results between local and central pathology laboratories, progression-free survival (PFS) and OS in HER2 positive patients were secondary objectives. Eligible patients had histologically proven locally advanced or metastatic gastric/GEJ cancer. HER2 testing was performed by means of IHC and in equivocal cases (IHC score 2+) in addition by ISH as a standard approach [21]. In the local laboratories, the HercepTest (Dako, Glostrup, Denmark), HER-2/neu (4B5) Rabbit Monoclonal Primary Antibody, PATHWAY, Roche (Ventana) and LDT c-erbB-2 Oncoprotein (HER-2), clone CB11 (Dako Autostainer Plus, Glostrup, Denmark) were used for IHC analysis, and ISH analysis was performed by using the PathVysion HER-2 DNA Probe Kit (Abbott, Des Plaines, USA), HER-2 Dual ISH DNA Probe Cocktail Assay, Roche (Ventana, Oro Valley, USA) or HER2 IQFISH pharmDx (Dako). In the central laboratory, HER2 status evaluation was performed by Prof. Peter Regitnig as single observer. Central IHC analysis was carried out using the HER-2/neu (4B5) Rabbit Monoclonal Primary Antibody, PATHWAY, Roche (Ventana) and interpretation was done according to the Rüschoff/Hofmann method [21]; ISH by brightfield HER-2 Dual ISH DNA Probe Cocktail assay, Roche (Ventana). Peter Regitnig received special training at Targos, Kassel, Germany, for HER2 testing in gastric cancer. Targos was the central laboratory for the ToGa trial [7]. Between January 2000 and December 2019, he performed 3668 HER2-IHC analyses and 895 HER2 ISH analyses. HER2 positive and HER2 negative samples were retrospectively sent to the central pathology laboratory for HER2 testing. At the time point of central HER2 testing, the central observer was not aware of the local HER2 test result. Due to the non-interventional character of this registry, central HER2 test results were not sent to the respective sites during the inclusion period of the GASTRIC-5 registry and therefore did not influence the treatment of individual patients during the course of disease. All HER2 positive patients received further therapy at the discretion of the local treating physician. Palliative systemic front-line therapy was initiated within a time frame of eight weeks after diagnosis of advanced gastric/GEJ cancer. 

### Statistical Analysis

Baseline characteristics were compared between HER2-negative and HER2-positive patients using crosstabulation together with the chi-squared test, in case of categorical data. Continuous data such as age were summarized using medians and ranges and compared between groups with Mann-Whitney test. 

The agreement of local and central testing was evaluated by calculating locally HER2 positive/centrally HER2 positive, locally HER2 negative/centrally HER2 negative, locally HER2 positive/centrally HER2 negative, and locally HER2 negative/centrally HER2 positive results. In addition, to describe year wise agreements between local and central testing, contingency coefficients were calculated for each year of testing. Kaplan–Meier survival curves together with log-rank testing were used to evaluate PFS and OS according to the diagnostic results. *p*-values < 0.05 were considered to indicate statistical significance. PFS was calculated from the date of diagnosis of metastatic disease until radiologically confirmed progression or death. Patients without progression at the last contact were censored. OS was calculated from the date of diagnosis of metastatic disease until death from any cause. Patients alive at the last contact were censored.

SPSS version 25 International Business Machines Corporation IBM was used for statistical analysis. R version 3.61 was used to calculate Kaplan-Meier survival curves.

## 3. Results

Between May 2011 and August 2018, 242 gastric/GEJ cancer patients were enrolled at nine sites in Austria. After exclusion of screening failures (*n* = 59), tumor samples of 183 patients were double tested for the HER2 status in the central pathology laboratory after initial local testing (Figure 1). The median time interval between local tissue sampling and dispatch for central HER2 status evaluation was 274 days. The respective HER2 test kits and application time frames are described in Appendix A.

### 3.1. Baseline Characteristics

According to central assessment, 38 out of 183 (21%) samples were considered HER2 positive. HER2 status analysis was based on biopsies and tissues obtained from definitive surgery in 133 (73%) and 50 (27%) cases, respectively. Baseline characteristics of the total population, HER2 negative and HER2 positive cohorts are shown in Table 1.

Low and intermediate grade samples (*p* < 0.001, Pearson chi-square) and the intestinal subtype according to Lauren’s classification (*p* < 0.001, Pearson chi-square) were significantly associated with HER2 positivity. Liver metastases (66% vs. 39%, *p* = 0.003, Pearson chi-square) and lung metastases (26% vs. 11%, *p* = 0.016, Pearson chi-square) were more frequently found in HER2 positive disease than in HER2 negative disease, whereas peritoneal spread showed a lower frequency (13% vs. 39%, *p* = 0.002, Pearson chi-square). 

The intestinal subtype showed a higher frequency of liver metastases (61% vs. 23%, *p* < 0.001, Pearson chi-square) and lung metastases (21% vs. 6%, *p* = 0.007, Pearson chi-square) compared to the diffuse/mixed subtype. Peritoneal metastases were more frequently found in the diffuse subtype in comparison to the intestinal subtype (53% vs. 20%, *p* < 0.001, Pearson chi-square).

### 3.2. HER2 Concordance and Discordance Rate

A discordant HER2 result between the local and central pathology laboratory was found in 22 (12%) cases with locally HER2 positive/centrally HER2 negative results (9%) exceeding locally HER2 negative/centrally HER2 positive results (3%) (Table 2). 

Neither tumor differentiation (low and intermediate grade: 18% vs. high grade: 11%, *p* = 0.315, Pearson chi-square), nor Lauren’s classification (intestinal: 14% versus diffuse and mixed: 9%, *p* = 0.525, Pearson chi-square) were associated with HER2 discordance rate. Discordant HER2 results were equally distributed between biopsy samples and tissue samples obtained from definite surgery (13% vs. 10%, *p* = 0.606, Pearson chi-square). HER2 concordance was more frequently found in “higher volume” (enrollment of > 20 patients) than in “lower volume” (enrollment of ≤ 20 patients) local pathology laboratories (92% versus 74%, *p* = 0.002, Pearson chi-square). The year of local HER2 testing did not significantly (*p* = 0.381) impact the HER2 concordance/discordance rate (Figure 2). Year-wise calculated contingency coefficients did not indicate a temporal trend regarding agreement between local and central HER2 testing (Figure 2). The time interval between local and central HER2 evaluation did not differ between discordant and concordant HER2 results (median: six months for both groups, *p* = 0.831 Mann-Whitney-*U* test).

### 3.3. HER2 Status Assessment by Immunohistochemistry (IHC)

The deviation of HER2 staining intensity between local and central testing is depicted in Appendix A. In cases of local HER2 IHC staining intensity exceeding central IHC staining intensity, a locally HER2 positive/centrally HER2 negative result was documented in 47% (Table 3).

### 3.4. HER2 Status Assessment by in Situ Hybridization (ISH)

HER2 ISH testing was performed in 27 (15%) and 20 (11%) samples in the local and central pathology laboratory, respectively. Despite an unequivocal negative HER2 test result based on IHC (score 0 or +1), an additional ISH test was carried out in one case and seven cases in the local and central pathology laboratory, respectively. However, each of these additional ISH tests confirmed HER2 negativity. 

Among local HER2 IHC 2+ samples (*n* = 26) the minority of cases (*n* = 10, 38%) proved to be HER2 amplified. The number of central IHC 2+ cases with HER2 amplification was even lower (two out of 13, 15%). A discordant ISH HER2 result was found in three out of six samples when ISH results were available from the local and central laboratory (Appendix A). 

### 3.5. Trastuzumab Based Palliative First-Line Therapy and Clinical Outcome

Twenty-eight patients with centrally confirmed HER2 positive disease received trastuzumab in combination with chemotherapy as palliative first-line therapy. The following chemotherapy backbones were applied: 5-FU plus a platinum (57%), 5-FU based triplet chemotherapy (28%), 5-FU (7%), 5-FU free doublet chemotherapy (4%), and a taxane (4%). A median duration of HER2-targeting therapy was 5.8 months (range: 0.3–63.8 months). Median follow-up has not been reached. HER2 targeting therapy was discontinued due to disease progression (68%), adverse events (14%), lost to follow-up (10%) or patient decision (4%) while therapy is still ongoing in one patient (4%).

Median PFS was 6.0 months (95% CI: 0.484–11.516, Figure 3a) and median OS was 17.7 months (95% CI: 10.870–24.530, Figure 3b). 

Patients with centrally confirmed HER2 positive disease receiving trastuzumab based front-line systemic therapy yielded the longest median OS (17.7 months; 95% CI: 10,870–24,530) of all groups (centrally HER2 positive without trastuzumab (locally HER2 negative or locally HER2 positive test results): 6.9 months, 95% CI: 3.980–9.820; locally HER2 positive/centrally HER2 negative with trastuzumab: 11.3 months, 95% CI: 5.364–17.236; centrally HER2 negative without trastuzumab: 12.0 months, 95% CI: 10,470–14,130, *p* = 0.019, Figure 4). Centrally HER2 positive patients treated without trastuzumab (locally HER2 negative or locally HER2 positive) showed a statistically significantly inferior survival in comparison to centrally confirmed HER2 positive trastuzumab treated patients (median OS: 6.9 months versus 17.7 months, *p* = 0.016, Figure 4).

## 4. Discussion

To the best of our knowledge, this is the first report of an investigation of HER2 positivity and HER2 concordance/discordance rates in a western real-world population with advanced gastric/GEJ cancer. We emphasize the non-interventional character of this registry: central HER2 test results were not sent to local sites and therefore did not influence treatment decisions of individual patients during the course of disease. The reported results of clinical outcome according to the HER2 status and according to the administration of trastuzumab highlight the absolute necessity of reproducibility of HER2 testing between various pathology laboratories. 

According to central assessment, 21% of patients tested HER2 positive and positive cases were associated with low- and intermediate grade histology as well as with the intestinal subtype according to Lauren’s classification as previously reported [7,20,22,23]. In contrast to the literature [24,25], HER2 positivity rates did not differ between gastric and GEJ cancer in the GASTRIC-5 registry. The distribution of metastases in the entire GASTRIC-5 cohort (liver > lung > peritoneum) is in line with data from the Swedish Cancer Registry [26]. The pattern of organ involvement was significantly associated with the HER2 status, favoring liver and lung metastases in HER2 positive disease but peritoneal spread in HER2 negative tumors (Table 1). However, the latter findings were mainly attributable to the underlying intestinal and diffuse/mixed (poorly cohesive) subtype according to Lauren’s classification and were less likely to be caused by the HER2 status.

Compared to the Asian HER-EAGLE study (3%) [20], the preliminary reports on the German VARIANZ study (23%) [27] and the French HERable study (9%) [28], we found a HER2 discordance rate of 12% between local and central assessment in an advanced western gastric/GEJ cancer cohort. However, the reported discordance rate in the GASTRIC-5 registry was based on local HER2 tests that had been performed over a period of ten years opposed to patient enrollment periods ranging from two to four years in the aforementioned studies. In locoregional breast cancer, discordant HER2 results were reported to be associated with lower pathological complete remission rates after neoadjuvant systemic therapy. However, the discordance rate between local and central HER2 testing has dramatically improved from 52% to 8% over a period of 12 years in breast cancer [29]. In the GASTRIC-5 registry we did not see a statistically significant learning effect of HER2 testing over time (Figure 2).

Intrapatient discordance rates between the primary tumor and metastases ranging from 2–24% [30,31,32] as well as intratumoral discordance rates [33,34] represent potential pitfalls of HER2 testing. Intratumoral HER2 expression heterogeneity can be found in up to 74% in early gastric cancer [35]. Neoplastic clonal selection with HER2 amplification in otherwise HER2 negative tumors as well as HER2 silenced tumor areas in cases with homogeneous HER2 amplification have been proposed as mechanisms leading to HER2 expression heterogeneity [36]. In consideration of HER2 expression heterogeneity between primary gastric/GEJ tumors and metastases as well as intratumoral HER2 expression heterogeneity, the application of anti-HER2 targeted therapy is markedly influenced by the analyzed tumor tissue/area. In order to circumvent intratumoral HER2 heterogeneity, at least five tumor-containing biopsies should be performed [37]. Furthermore, concomitant assessment of HER2 status based on the primary tumor and synchronous metastases tissue may circumvent intrapatient HER2 heterogeneity [38]. Re-biopsies are recommended in initially HER2 negative tumors in case of recurrence [37,39].

Deeper sections or even different tumor blocks were centrally re-tested, which could have had an influence on the divergent results due to the known intratumoral heterogeneity of HER2 positivity. Non-specific staining in the marginated cytoplasm is another cause of HER2 status misinterpretation in gastric cancer with signet ring cell histology [40], a subtype, which was not specifically assessed within this registry. However, signet ring cell histology is overlapping with the diffuse subtype according to Lauren’s classification [41], which had no impact on HER2 discordance rates in our study.

While locally HER2 negative/centrally HER2 positive results exceeded locally HER2 positive/centrally HER2 negative results in the HERable study [28] and HER-EAGLE study [20], the opposite was the case in our study (3% vs. 9%, Table 2). In the GASTRIC-5 registry, local pathology laboratories were more likely to report higher HER2 IHC scores, thereby increasing locally HER2 positive/centrally HER2 negative cases (Table 3). ISH provides higher accuracy in comparison to IHC when assessing the HER2 status in gastric/GEJ cancer [7,30] and therefore ISH can be considered as the reference method. However, against expectations, the discordance rate of HER2 results between local and central assessment by ISH was 50% (3/6) (Appendix A). Despite small numbers, our ISH-based HER2 disagreement rate was considerably higher than in the TRIO-013/LOGiC trial (5%) [8], which compared different FISH assay methods between central laboratories. We cannot rule out that the heterogeneity of applied HER2 test kits used at local pathology laboratories may have contributed to discordant HER2 results (Appendix A). Evidently, only IVD-CE certified and/or FDA approved HER2 test kits should be used for this crucial test.

Local and central HER2 IHC 2+ samples turned out be HER2 amplified in only 38% and 15% of cases, respectively. In contrast to our findings, the majority (84%) of IHC 2+ samples showed a HER2 amplification in the TRIO-013/LOGiC trial [8]. Performing additional ISH in unequivocal HER2 results did neither affect HER2 results nor clinical decision-making in individual cases and cannot be recommended in clinical practice.

Among the centers participating in the GASTRIC-5 registry, local pathology centers enrolling more than 20 patients achieved better reproducibility of HER2 results, which corroborates the necessity to define a minimum number of annual HER2 assessments at local pathology laboratories or alternatively central HER2 testing at reference laboratories.

Median OS among centrally confirmed HER2 positive patients undergoing trastuzumab based therapy was encouraging with 17.7 months outside a clinical trial, when compared to the ToGA trial with a median OS of 16.0 months in patients with IHC 2+ and FISH-positive tumors or IHC 3+ tumors [7]. The vast majority of patients (85%) received a 5-FU plus platinum chemotherapy backbone in analogy to the ToGA trial as either doublet or triplet chemotherapy in combination with trastuzumab; therefore, comparison of clinical outcome is admissible. Although data on subsequent therapy protocols are not available within the GASTRIC-5 registry, we assume that the availability of ramucirumab [42,43], nivolumab [44], pembrolizumab [45] and TAS-102 [46] within the enrollment period from 2011 to 2018 has influenced the favorable clinical outcome of the trastuzumab treated centrally confirmed HER2 positive cohort. Trastuzumab was combined with either FLOT (5-FU, leucovorin, oxaliplatin, docetaxel) or EOX (epirubicin, oxaliplatin, capecitabine) triplet chemotherapy in one in four HER2 positive patients. On the one hand, triplet chemotherapy with 5-FU, leucovorin, oxaliplatin and docetaxel (FLOT or DCF protocol) may improve OS when compared to doublet chemotherapy [12]; on the other hand, the benefit of epirubicin within triplet-chemotherapy protocols such as EOX is highly doubtful in gastric/GEJ cancer [47]. In consideration of a subsidiary role of anthracyclines in the treatment of gastric/GEJ cancer and the fact that several patients (11%) only received monochemotherapy in combination with trastuzumab, we believe that the encouraging clinical outcome of the centrally confirmed HER2 positive cohort treated with anti-HER2 targeting therapy was independent of triplet chemotherapy backbones [48]. Centrally HER2 positive patients not receiving trastuzumab had the worst outcome with a median OS of 6.9 months (Figure 4), a finding which is in line with previous studies [7,27].

In conclusion, the HER2 positivity rate of 21% in this real-world advanced western gastric/GEJ cancer cohort was comparable to reported rates in clinical phase III trials [7,15]. Due to the clinically meaningful survival benefit of adding trastuzumab to first-line systemic therapy in centrally confirmed HER2 positive advanced gastric/GEJ cancer, minimizing discordant HER2 results, especially locally HER2 negative/centrally HER2 positive results, is an absolute necessity. In clinical practice, the latter goal may be achieved by sending gastric/GEJ cancer samples to higher volume pathology centers and/or by applying the respective IVD-CE certified and/or FDA approved HER2 companion tests.

## Figures and Tables

**Figure 1 jcm-09-00935-f001:**
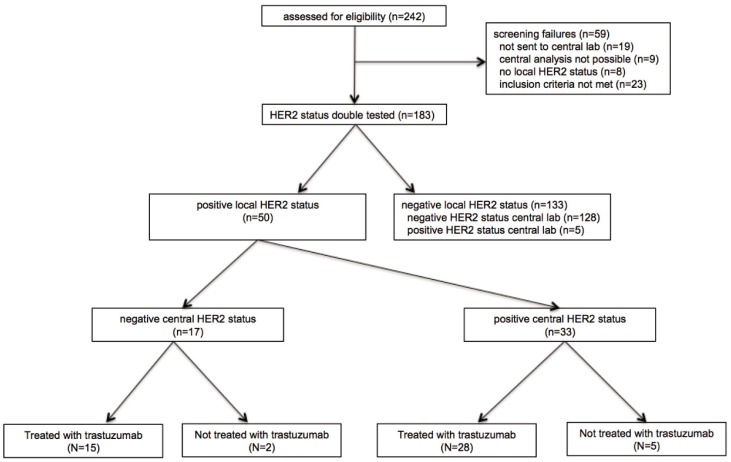
Consort diagram GASTRIC-5 registry.

**Figure 2 jcm-09-00935-f002:**
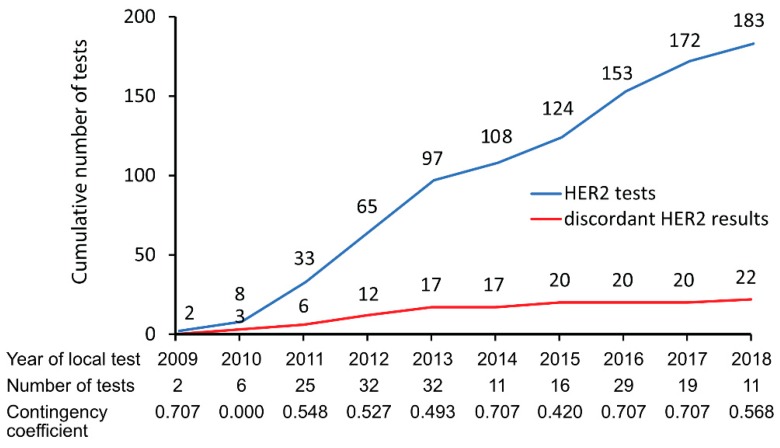
HER2 discordance rate according to local HER2 testing year. Blue line: cumulative HER2 tests, red line: cumulative discordant HER2 results.

**Figure 3 jcm-09-00935-f003:**
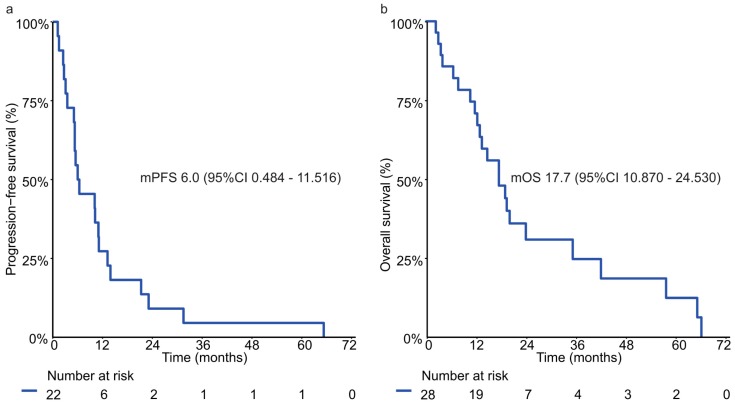
Progression-free survival (**a**) and overall survival (**b**) with palliative first-line trastuzumab based systemic therapy in centrally HER2 positive advanced gastric/GEJ cancer patients. The tick marks on the curves represents censored patients.

**Figure 4 jcm-09-00935-f004:**
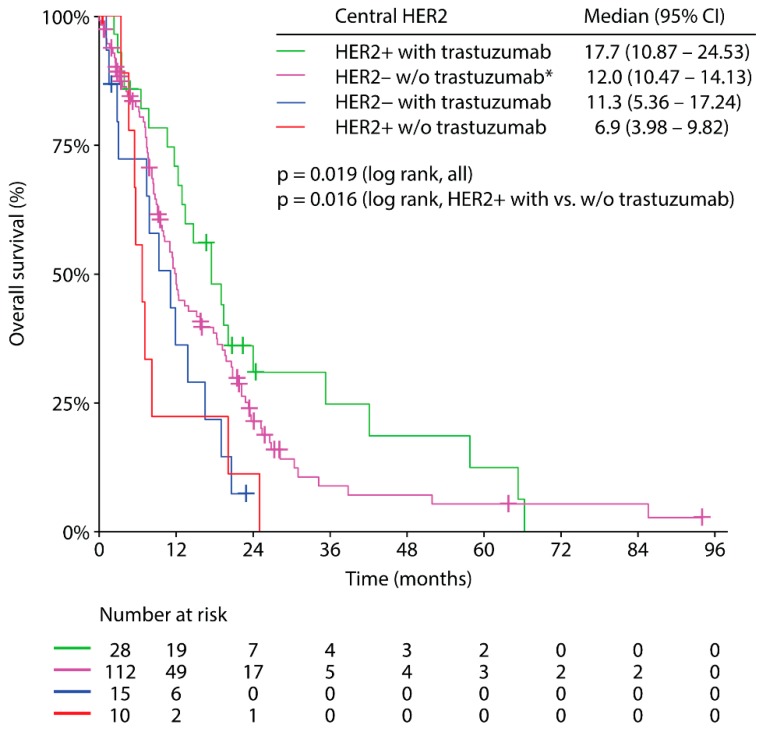
Overall survival from diagnosis of metastatic gastric/GEJ cancer according to central HER2 status and according to trastuzumab front-line treatment status. The tick marks on the curves represents censored patients. * According to standard of care.

**Table 1 jcm-09-00935-t001:** Baseline characteristics according to central HER2 status.

	All Patients *n* = 183 (100%)	HER2 Negative *n* = 145 (79%)	HER2 Positive *n* = 38 (21%)	*p*-Value
Age (median)	67	67	66	0.378 *
Range	28–89	28–89	45–86	
**Sex**				0.922
Male	124 (68)	98 (68)	26 (68)	
Female	59 (32)	47 (32)	12 (32)	
**Primary Tumor**				0.403
Gastric cancer	93 (55)	75 (56)	18 (49)	
GEJ cancer	77 (45)	58 (44)	19 (51)	
NA	13	12	1	
**Prior Surgery**				0.211
Yes	74 (40)	62 (43)	12 (32)	
No	109 (60)	83 (57)	26 (68)	
**Histology Grading**				<0.001
1	3 (2)	2 (2)	1 (3)	
2	48 (33)	27 (24)	21 (60)	
3	94 (65)	81 (74)	13 (37)	
NA	38	35	3	
**Lauren’s Classification**				<0.001
Intestinal	86 (55)	57 (46)	29 (88)	
Diffuse	60 (39)	57 (46)	3 (9)	
Mixed	10 (6)	9 (8)	1 (3)	
NA	27	22	5	
**Distribution of Metastases**				
Liver ^a^	81 (44)	56 (39)	25 (66)	0.003
Peritoneum ^a^	62 (34)	57 (39)	5 (13)	0.002
Lung ^a^	26 (14)	16 (11)	10 (26)	0.016
Distant lymph nodes ^a^	25 (14)	20 (14)	5 (13)	0.919
Other ^a^	15 (8)	13 (9)	2 (5)	0.459

* Mann-Whitney-*U*-test; ^a^ multiple selection possible; GEJ: gastroesophageal junction, HER2: human epidermal growth factor receptor 2, NA: not available.

**Table 2 jcm-09-00935-t002:** HER2 concordance/discordance rate according to local/central testing.

		Local HER2 Test	
		Negative	Positive
**Central HER2 Test**	Negative	128 (70%)	17 (9%)
	Positive	5 (3%)	33 (18%)

**Table 3 jcm-09-00935-t003:** Comparison of local and central HER2 IHC result and impact on HER2 status.

	HER2 Result (%)			
	Locally HER2 Negative/ Centrally HER2 Negative	Locally HER2 Positive/ Centrally HER2 Positive	Locally HER2 Negative/ Centrally HER2 Positive	Locally HER2 Positive/ Centrally HER2 Negative
Local IHC < central IHC	56 (90)	1 (2)	5 (8)	0 (0)
Local IHC > central IHC	17 (50)	1 (3)	0 (0)	16 (47)
Local IHC = central IHC	55 (63)	31 (36)	0 (0)	1 (1)
Total	128	33	5	17

IHC: immunohistochemistry.

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
