# Peer review of "Local and Central Evaluation of HER2 Positivity and Clinical Outcome in Advanced Gastric and Gastroesophageal Cancer—Results from the AGMT GASTRIC-5 Registry"

_jcm, 2020, doi:10.3390/jcm9040935_

Round 1
Reviewer #1 point #1:
„It is well known that evaluation of HER2 status represents a crucial point in the management of patients affected by gastric and/or GEJ cancer. In particular, the pathological assessment of the HER2 raised some difficulties and discrepancies in many countries. In the present study, a cohort of Austrian pathologists utilized the GASTRIC-5 registry as an observational, multi-center instrument to compare local and central HER2 testing. As usual, HER2 status was firstly assessed by immunohistochemistry (IHC) and in equivocal cases (IHC score 2+) by additional in-situ hybridization. Discordant HER2 results were found in 12% (22/183) with locally HER2 positive/centrally HER2 negative results (9%, 17 out of 183) exceeding locally HER2 negative/centrally HER2 positive results (3%, 5/183). Nevertheless, immunohistochemical analyses have been performed in a not uniform homogeneous manner, since 7 sites utilized HER-2/neu (4B5) Rabbit Monoclonal Primary Antibody, PATHWAY; Roche Ventana, 2 sites applied HerceptTest™; DAKO and 1 site LDT c-erbB-2 Oncoprotein (HER-2), clone CB11 stained on DAKO Autostainer Plus; therefore, on the light of this procedure, it is not surprising to obtain some discordant results due to different sensibility and specificity of utilized antibodies. However, also for in situ hybridization methodologies, 3 sites used 2+ Ventana Inform HER-2 Dual ISH DNA Probe Cocktail Assay (Roche), 3 sites applied PathVysion HER-2 DNA Probe Kit (Abbott) and finally 1 site utilized HER2 IQFISH pharmDx - (Dako Agilent). Consequently, although the findings obtained by the GASTRIC-5 registry corroborate the necessity for central quality control to optimize HER2 pathological assessment as well as the consequent personalized treatment, I will suggest to perform again the analysis (local versus central), applying the same methodology in order to strictly verify the existence of such discordant cases.“
Authors’ response:
We agree with the reviewer’s comment concerning the heterogeneity of applied HER2 test kits. However, this issue has been clearly stated in the “Discussion” section (page 9, line 310-312). Although it would be interesting to uniformly assess local HER2 status, such an approach is not feasible within this registry design and tumor material has been largely consumed in the majority of cases.
Reviewer #1 point #2:
„Moreover, some comments concerning the intratumoral HER2 heterogeneity as well as the discordant HER2 status between primary gastric carcinomas and corresponding lymph node or distant metastases should be added and discussed, taking also in consideration data elsewhere published on the same topic:
- Intratumoral HER2 heterogeneity in early gastric carcinomas: potential bias in therapeutic management. Virchows Arch. 2019; 474:401-402
- Discordance Rate of HER2 Status in Primary Gastric Cancer and Synchronous Lymph Node Metastases: Its Impact on Therapeutic Decision and Clinical Management. Pathol Oncol Res. 2018;24:695-696.“
Authors’ response:
As proposed by reviewer #1, intratumoral HER2 heterogeneity as well as intrapatient HER2 heterogeneity have been discussed in more detail in the „Discussion“ section with reference to the respective literature (page 9, line 282-292).
Reviewer 2 Report
Reviewer #2:
„I am agree that it is the first report of an investigation of HER2 positivity and HER2 concordance/discordance rates in a western real-world population with advanced gastric/GEJ cancer. The study of Austrian Group for Medical Tumor Therapy (AGMT) concerns an observational, multi-center research in Austria. The study concerns verification of patients with locally advanced or metastatic gastric/GEJ cancer for treatment of Trastuzumab. The patients were tested for the HER2 status in the central pathology laboratory after initial local testing. In the central laboratory HER2 status evaluation was performed by single observer. The authors discuss the intrapatient discordance rates between the primary tumor and metastases ranging from 2-24% as well as intratumoral discordance rates which represent potential pitfalls of HER2 testing. Besides, the interpretation of HER2 immunohistochemical reaction is a problem. HER2 IHC and ISH require reproducibility of HER2 results. The work carefully discusses these difficult problems. The authors analyze and discuss current literature. In summary, the role of the pathologist is crucial. It is very important proposal for clinical practice, to send gastric/GEJ cancer samples to higher volume pathology centers and/or by applying the respective IVD-CE certified and/or FDA approved HER2 312 companion tests.“
Authors’ response:
The reviewer did not raise any issue.